# IA-RED$^2$: Interpretability-Aware Redundancy Reduction for Vision Transformers

**Bowen Pan**[1], **Rameswar Panda**[2], **Yifan Jiang**[3], **Zhangyang Wang**[3], **Rogerio Feris**[2], **Aude Oliva**[1,2]

[1]MIT CSAIL, [2]MIT-IBM Watson AI Lab, [3]UT Austin

## Abstract

The self-attention-based model, transformer, is recently becoming the leading backbone in the field of computer vision. In spite of the impressive success made by transformers in a variety of vision tasks, it still suffers from heavy computation and intensive memory costs. To address this limitation, this paper presents an Interpretability-Aware REDundancy REDuction framework (IA-RED$^2$). We start by observing a large amount of redundant computation, mainly spent on uncorrelated input patches, and then introduce an interpretable module to dynamically and gracefully drop these redundant patches. This novel framework is then extended to a hierarchical structure, where uncorrelated tokens at different stages are gradually removed, resulting in a considerable shrinkage of computational cost. We include extensive experiments on both image and video tasks, where our method could deliver up to **1.4× speed-up** for state-of-the-art models like DeiT [53] and TimeSformer [3], by only sacrificing **less than 0.7%** accuracy. More importantly, contrary to other acceleration approaches, our method is **inherently interpretable** with substantial visual evidence, making vision transformer closer to a more human-understandable architecture while being lighter. We demonstrate that the interpretability that naturally emerged in our framework can outperform the raw attention learned by the original visual transformer, as well as those generated by off-the-shelf interpretation methods, with both qualitative and quantitative results. Project Page: `http://people.csail.mit.edu/bpan/ia-red/`.

## 1 Introduction

Transformer, a self-attention-based architecture processing sequential input without any recurrent or convolutional operations, has set off a storm in the computer vision literature recently. By dividing the input image into a series of patches and then tokenizing them with linear transformation, the transformer can effectively process the visual data in different modalities [13, 53, 54, 28, 3, 17, 66]. Despite its versatility, the transformer is always deeply troubled with inefficient computation and its vague interpretability. The vision transformer suffers heavy computational costs, especially when the input sequence is long. As the attention module in the vision transformer computes the fully-connected relations among all of the input patches, the computational cost is then quadratic with regard to the length of the input sequence. On the other hand, previous works [6, 8] have already shown the vulnerable interpretability of the original vision transformer, where the raw attention comes from the architecture sometimes fails to perceive the informative region of the input images.

Recently, more designs of vision transformer architecture [34, 65, 18, 56, 14, 9, 3] are proposed to get higher accuracy with less computational cost. Although these methods anchor good trade-offs between efficiency and accuracy, their compression makes the vision transformer even more lack interpretability. Most of these methods assume that the input sequences are sampled from a regular visual input in a fixed shape rule, and thus their network architectures are not flexible as well, which makes the vision transformer (1) no longer able to process the input sequence with arbitrary length as the architecture is designed for a specific input shape; (2) neither model-agnostic nor task agnostic anymore; or (3) neglect the fact that the model redundancy is also input-dependant. We yet argue that

35th Conference on Neural Information Processing Systems (NeurIPS 2021).

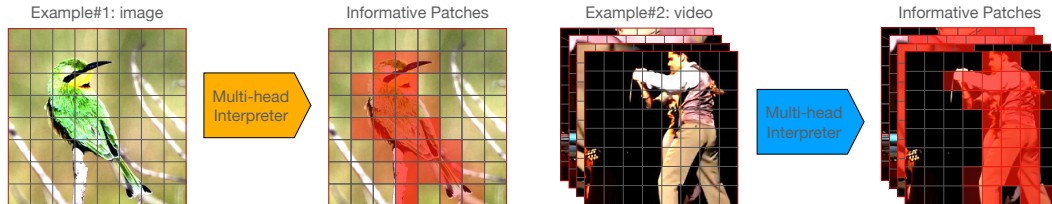

Figure 1: Two examples of redundancy reduction in vision transformers. Our proposed multi-head interpreters serve as a model-agnostic module which are built on top of the existing transformer-based backbones for different tasks, including image recognition and video action recognition.

there is **no inherent tension** between efficiency and interpretability, and achieving them both does **not** have to pay design flexibility as a price. Indeed, starting from the philosophy of *Occam's razor*, the law of parsimony, or always pursuing more compact solutions when possible, is always treated as a rule-of-thumb for pursing interpretability, especially in complicated fitting problems [21].

This paper aims to seek the win-win between efficiency and interpretability while keeping the flexibility and versatility of the original vision transformer. We propose a novel Interpretability-Aware REDundancy REDuction (IA-RED$^2$) framework for reducing the redundancy of vision transformers. The key mechanism that IA-RED$^2$ uses to increase efficiency is to dynamically drop some less informative patches in the original input sequence so that the length of the input sequence could be reduced. While the original vision transformer tokenizes all of the input patches, it neglects the fact that some of the input patches are redundant and such redundancy is input-dependant (see from Figure 1). As the computational complexity of the attention module is quadratically linear to the input sequence length, the effect of reducing input sequence length would be magnified in the amount of the computation. Motivated by this, we leverage the idea of dynamic inference [39, 37, 38, 61, 57], and adopt a policy network (referred to as *multi-head interpreter*) to decide which patches are uninformative and then discard them. Our proposed method is inherently interpretability-aware as the policy network learns to discriminate which region is crucial for the final prediction results.

To summarize, the main contributions of our work includes: (1) We propose IA-RED$^2$, the first **interpretability-aware** redundancy reduction framework for vision transformer. (2) Our IA-RED$^2$ framework is **one of the first input-dependent** dynamic inference framework for vision transformer, which adaptively decides the patch tokens to compute per input instance. (3) IA-RED$^2$ is both **model-agnostic and task-agnostic**. We conduct experiments with IA-RED$^2$ framework spanning different tasks, including image recognition and action recognition, and different models, including DeiT [53], TimeSformer [3]. (4) We attain **promising interpretable results** (shown in Figure 3) over baselines, with a 1.4× acceleration over DeiT on image recognition tasks, and a 4× acceleration over TimeSformer on video action recognition task while largely maintaining the accuracy. We also provide both qualitative results regarding interpretability with heatmaps by our method and those from other baseline methods like raw attention, MemNet [29]; as well as the quantitative comparison with current state-of-the-art model interpretability methods, such as GradCAM [44], on ImageNet-Segmentation [16] dataset with the weakly-supervised image segmentation task.

## 2 Related Work

**Interpretability of Neural Networks.** Besides improving the discrimination power of deep neural networks, model interpretability has recently raised another significant and popular research question. One of the important goals is to predict the heatmap visualization that precisely indicates the objects or contexts of relevance. Simonyan *et. al.* [45] attempts to maximize the class score that generates a saliency map for the given inputs. Dabkowski *et al.* [12] mask the salient parts of the inputs to manipulate the scores of the classifier, which generalizes well to unseen images and enables fast saliency detection. Khosla *et al.* [29] provide the largest annotated image memorability dataset to benchmark the visualization and explanation of natural images. After that, gradient-based methods [49, 46, 47] are proposed to generate precise heatmaps, by computing the gradient with respect to input during the backpropagation. While all the above approaches are studying the interpretability of convolutional neural networks (CNNs), only a few works contribute to the visualization of the vision transformer. Chefer *et al.* shed some light on its visualization by assigning the local relevance on Transformer layers. Caron *et al.* [6] demonstrate that a self-supervised trained ViT produces explicit representation about the semantic location of a given object in natural images. Different from all of them, our approach starts with a novel multi-head interpreter which is supervised

by an efficiency-driven signal, and then benefits from this powerful interpreter by reducing the redundancy of transformer, achieving a "win-win" between interpretability and efficiency.

**Dynamic Networks.** Neural networks are found as redundant regarding their huge computation cost [19, 23, 35]. To overcame this issue, many adaptive computation methods are explored during the inference stage [1, 2, 57, 15, 26, 58, 20]. These adaptive computation strategies help speed up the inference time of convolutional neural networks (CNNs) [33], recurrent neural network (RNNs) [15], and also self-attention based methods (BERT) [25]. Besides the model-level adaptation, others further extend this idea to data-level adaptation, by either reducing the spatial redundancy [63] or focusing on key area [59]. However, those methods are limited by the convolutional structure, where only 2D data can be taken as input. Different from those approaches, our methods naturally benefit from the unstructured input taken by vision transformer, and thus can provide a much more precise glance at the target object with the affection of background being eliminated.

**Vision Transformer.** Transformer, as a self-attention based model, has been widely adopted in natural language processing area before. The recent advance [13] shows that the transformer can also achieve incredible performance on computer vision tasks. While vision transformer suffers from the necessity large-scaled dataset [48], many recent works try to encode strong inductive prior by either combining it with convolutional layer [60, 32, 64, 62] or introducing 2D-hierarchical structure to vision transformer [34, 56, 14, 9]. Besides, transformer also shows strong power in other vision tasks, including semantic segmentation [67], object detection [5, 68], image processing [10], and image generation [28, 27]. These successes further suggest the potential of transformer to become the universal model for general vision tasks. Some other works [40, 11, 51] also make meaningful efforts on vision transformer efficiency. Different from those methods, the proposed method achieves a "win-win" on both efficiency and interpretability.

## 3 Proposed Method

Our main goal is to reduce the redundancy in vision transformers by dynamically dropping less informative patches in the original input sequence while classifying it correctly with the minimum computation. Our method is built on top of vision transformer (ViT) [13]. We start from presenting a brief overview of ViT, including the computational complexity of each module regarding the input sequence length. We then describe our proposed IA-RED$^2$ framework for hierarchically reducing the redundant patch tokens at different layers of the vision transformer.

### 3.1 Overview of Vision Transformer

Vision transformer mainly consists of three main modules: (1) Multi-head Self Attention layer (MSA) to learn relationships between every two different patches among all the input tokens. There are $h$ self-attention heads inside the MSA. In each self-attention head, the input token $X_i$ is first projected to a query $Q_i$, a key $K_i$, and a value $V_i$ by three different linear transformations. Then, the query $Q_i$ computes the dot products with all the keys $K$ and these dot products will be scaled and normalized by the softmax layer to get the attention weights. After that, it outputs the token $Y_i$ by weighted sum up all the values $V$ with the obtained attention weights. Finally, the outputs from all heads are concatenated and re-projected by a linear layer into an output token. (2) Feed-Forward Network (FFN) which consists of two linear layers which are connected by the GeLU activation [24] function. For each output token $Y_i \in R^D$ from the precedent MSA layer, FFN processes it individually. The first linear layer upgrades its dimension from $D$ to $4D$, and the second linear layer downgrades its dimension from $4D$ to $D$. Both MSA and FFN are functioning as residual connection [22]. (3) Linear Patch Embedding and Positional Encoding: For an image or a video clip, ViT first splits it into several fixed-size patches and embeds them into input tokens with a linear layer. After transforming the original image and video into a series of tokens, the network is no longer capable of being aware of the positional information of the input tokens. Thus the positional embeddings are added to the input tokens right after the patch embedding to learn the positional information of each token.

**Computational Complexity.** For an input sequence $N \times D$, where $N$ is the length of the input sequence and $D$ is the embedding dimension of each input token. The computation complexity of the MSA is $\mathcal{O}(4ND^2 + 2N^2D)$. While for the FFN, the computational complexity is $\mathcal{O}(8ND^2)$. As the computational complexity of patch embedding can be neglected compared with the MSA and FFN, the total computational complexity of the ViT is $\mathcal{O}(12ND^2 + 2N^2D)$.

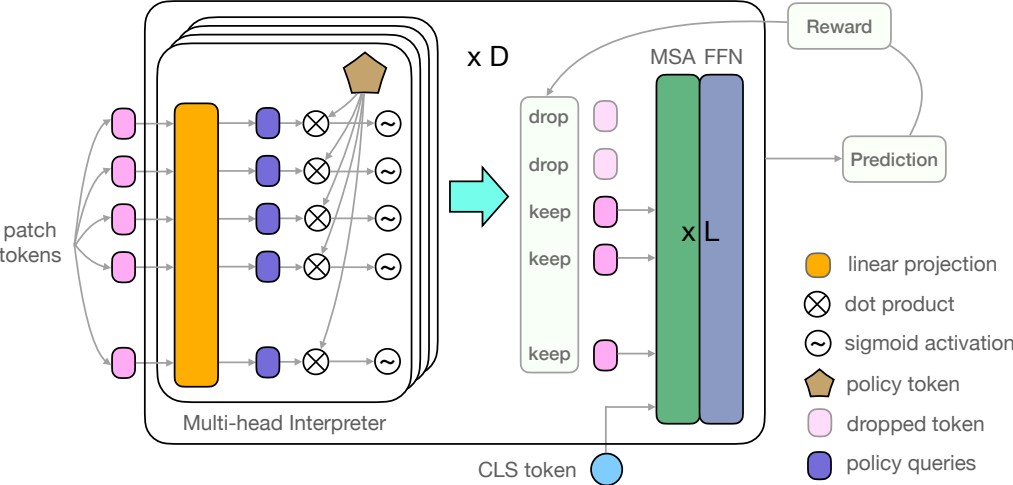

Figure 2: Illustration of our proposed IA-RED$^2$ framework. We divide the transformer into $D$ groups. Each group contains a multi-head interpreter and $L$ combinations of the MSA and FFN. Before input to the MSA and FFN, the patch tokens will be evaluated by the multi-head interpreter to drop some uninformative patches. The multi-head interpreters are optimized by reward considering both the efficiency and accuracy. Best viewed in color.

## 3.2 Interpretability-Aware Redundancy Reduction

In this section, we introduce our multi-head interpreter in detail which uses a policy token to estimate the importance of the input token. We also demonstrate how we hierarchically train the multi-head interpreter based on a pre-trained vision transformer. Finally, we illustrate that how the interpretability emerges in our IA-RED$^2$ framework.

**Multi-head Interpreter.** We borrow the idea from the architecture of the MSA layer to devise our policy module, named multi-head interpreter. Given a sequence of patch tokens $X \in R^{N \times d}$ which already contain the positional information, we drop the uninformative patch tokens by using the multi-head interpreter. We first divide the original ViT evenly into $D$ groups, each group contains a multi-head interpreter and $L$ blocks which consists of one MSA layer and one FFN. Inside each group, before inputting to the blocks, the patch tokens will first be evaluated by the multi-head interpreter for the informative score $I_{ij}$, where $i$ and $j$ represent the position of the input token and the group respectively. If $I_{ij}$ is below the threshold 0.5, the patch $X_i$ will be completely discarded at $j^{th}$ group and will not be available in the subsequent groups. The $I_{ij}$ is obtained by:

$$I_{ij} = \frac{1}{H} \sum_h \phi(\mathcal{F}_q^h(X_i) * \mathcal{F}_k^h(P_j)), \tag{1}$$

where $P_j$ is the policy token in the $j^{th}$ multi-head interpreter, $H$ is the number of the heads in the multi-head interpreter, $\mathcal{F}_q^h$ and $\mathcal{F}_k^h$ are the linear layer at $h^{th}$ head for the patch tokens and the policy token respectively, $*$ represents the dot product and $\phi$ the sigmoid activation function.

**Hierarchical Training Scheme.** Our hierarchical training scheme is built on top of a well-trained ViT. In our IA-RED$^2$ framework, all of the MSA-FFN blocks in the original vision transformer will be evenly assigned into D groups in our IA-RED$^2$ framework, where each group contains L MSA-FFN blocks and one multi-head interpreter. We fix the parameters of the patch embedding layer, positional encoding, and the class token during the training, and only focus on the parameters inside each group. The network groups are optimized in a curriculum learning manner. For example, if the number of groups D is 3, we will first optimize groups 1 to 3, then 2 to 3, and finally, we optimize the third group. Intuitively, we hope the interpreter at the early stage could learn to select the patches containing all of the necessary contextual information for the correct final prediction, while the interpreter at later stages could focus more on the part-level information since now each token's information has already gone through global interaction and fusion. The pseudo-code for the above optimization pipeline can be referred in supplementary materials. We optimize the multi-head interpreters by using the REINFORCE method where the reward considers both the efficiency and accuracy, and finetune the MSA-FFN blocks with gradients computed based on cross-entropy loss.

Formally, during the training phase, given a sequence of patch tokens $X \in R^{N \times d}$ input to the $j^{th}$ multi-head interpreter, the multi-head interpreter will generate policies for each input token of dropping or keeping it as Bernoulli distribution by: $\pi_W(u_i|X_i) = I_{ij}^{u_i} * (1 - I_{ij})^{1-u_i}$, where $u_i = 1$ means to keep the token and $u_i = 0$ means to discard the token, $I_{ij}$ is defined in the Eq. 1 and $X_i$ denotes the $i^{th}$ token in the token sequence $X$. We associate these actions with the reward function:

$$R(u) = \begin{cases} 1 - (\frac{|u|_0}{N})^2 & \text{if correct} \\ -\tau & \text{otherwise} \end{cases}, \tag{2}$$

where $(\frac{|u|_0}{N})^2$ measures the percentage of the patches kept, and $\tau$ is the value of penalty for the error prediction which controls the trade-off between the efficiency and the accuracy of the network. This reward function encourages the multi-head interpreter to predict the correct results with as few patch tokens as possible. Then we optimize the multi-head interpreter individually by the expected gradient:

$$\nabla_{W_j} J = E_{u \sim \pi} [A \nabla_{W_j} \sum_{i=1}^{N} log[I_{ij} u_i + (1 - I_{ij})(1 - u_i)]], \quad A = R(u) - R(\hat{u}), \tag{3}$$

where $J = E_{u \sim \pi}[R(u)]$ is the expected reward to compute the policy gradient [50], $W_j$ denotes the parameters of the $j^{th}$ multi-head interpreter. We use the self-critical baseline $R(\hat{u})$ in [41] to reduce the variance of optimization, where $\hat{u}$ denotes the maximally probable configuration under the current policy: i.e., $u_i = 1$ if $I_{ij} > 0.5$, and $u_i = 0$ otherwise. As the computation of the $j^{th}$ multi-head interpreter is based on the output tokens of $(j-1)^{th}$ group, we optimize the entire network in a curriculum learning manner. We first train the interpreter in the earlier layer, and then fix the interpreter and finetune all of the subsequent MSA-FFN blocks. Let's take the $j^{th}$ group for example. For the $j^{th}$ group, we first only train the multi-head interpreter and then fix it while optimizing the subsequent MSA-FFN modules in the $j^{th}$, ... , $D^{th}$ groups. When we optimize the $j^{th}$ group, the multi-head interpreter in the latter groups will be masked and keep all of the tokens.

**Emergence of Interpretability.** By visualizing the informative scores predicted by the multi-head interpreters in different network groups, we can see the redundancy of the input patches is hierarchically reduced at different levels clearly. For those patches that are removed in the precedent groups, we treat the informative score of them as zero. Thus we can obtain a sequence of the informative scores from each network group whose length equals the original input sequence length. We rearrange this score sequence and interpolate it back to the size of the input vision content (e.g. image or video). As the range of the informative score is from 0 to 1, we can draw a heatmap for each network group which interprets that what is redundant for this network group.

## 4 Experiments

**Datasets and Metrics.** We conduct image recognition experiments on the ImageNet-1k classification dataset [31]. The performance of our models on ImageNet-1k is measured with the metrics of top-1 and top-5 accuracy rates. For weakly-supervised image segmentation experiments, we adopt the ImageNet-Segmentation dataset [16] to evaluate the heatmaps we generate. We report three metrics: pixel accuracy, mean accuracy (mAcc), and mean IoU (mIoU) to reflect the segmentation performance. Finally, for video action recognition, we conduct our experiments on Kinetics-400 dataset [7], which contains 240k training videos and 10K videos for testing across 400 classes. We report the metrics of clip-1 and video-1 error of video models, which denotes the error rate of evaluating the model with the single clip and the Left-Center-Right three clips, respectively.

**Model Architectures.** We build our image model on top of DeiT [53] which adopts the architecture of the ViT [13] by modifying the depth and width. Compared to the original ViT [13], DeiT has a distillation token that is in charge of distilling the knowledge from the teacher CNN network. DeiT is trained and evaluated on ImageNet-1k [31], without large-scale pre-training. We choose DeiT-S and DeiT-B as our base models, where DeiT-B is $4\times$ larger than DeiT-S in terms of FLOPs. For the video model, we construct our model based on TimeSformer [3]. There are several different attention mechanisms introduced in [3]. Here we adopt the TimeSformer with the JointST attention method, which keeps the architecture of the vanilla ViT and takes all of the input patches as one sequence. Our model samples 8 frames in one video clip and splits them into 1568 frame patches. During inference, our model evenly crops 3 views from the video clip, each view of them has 8 frames.

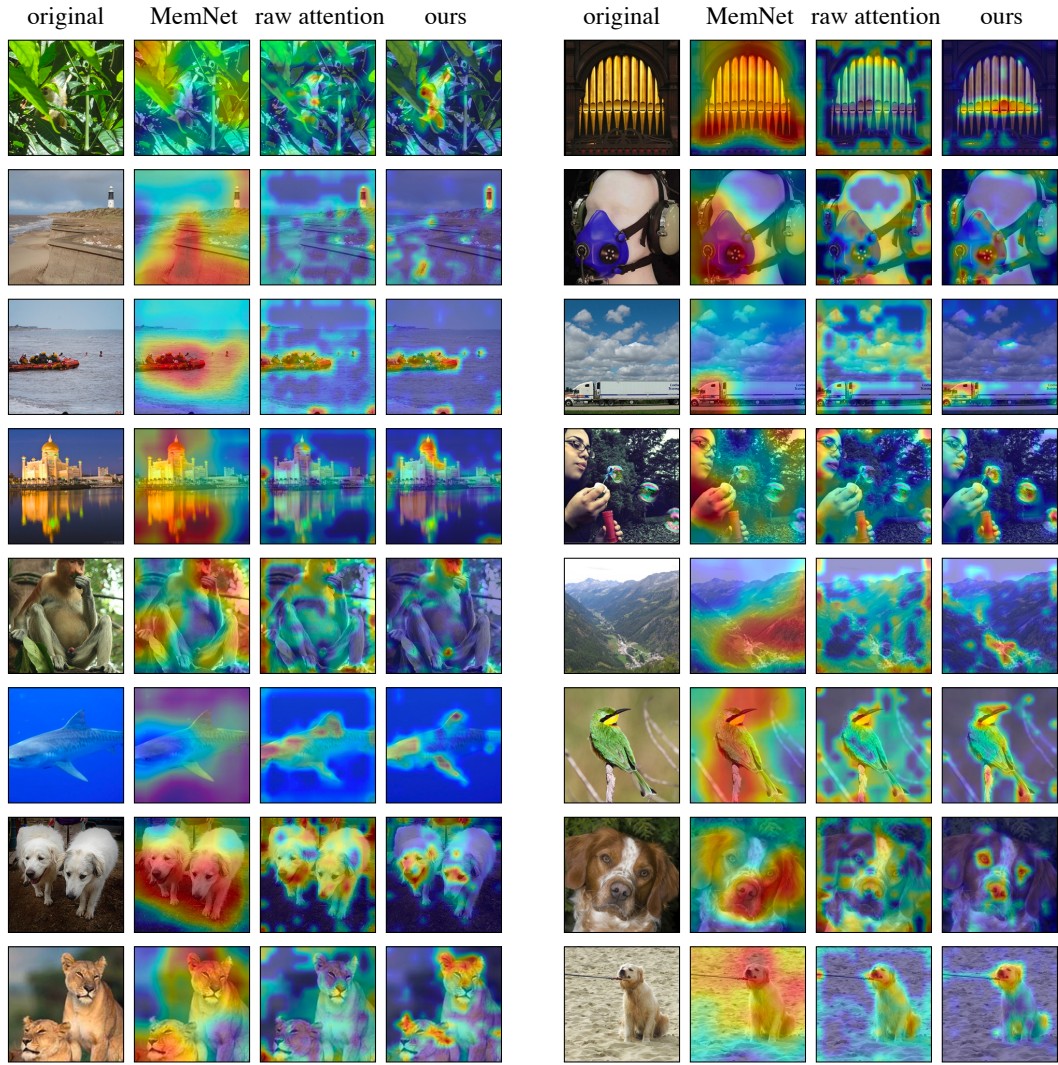

| original | MemNet | raw attention | ours | | original | MemNet | raw attention | ours |

Figure 3: We visualize the heatmaps which highlight the informative region of the input images of MemNet, raw attention at the second block, and our method with DeiT-S model. We find that our method can obviously better interpret the part-level stuff of the objects of interest. Here the visualization results are randomly chosen. Best viewed in color.

Table 1: Results of weakly-supervised image segmentation on ImageNet-segmentation [16]. We use our method based on the training with DeiT-S model. Higher is better.

| Metrics | raw attention | LIME [42] | MemNet [29] | GradCAM [44] | LRP [4] | Ours |
|---|---|---|---|---|---|---|
| pixel accuracy | 67.87 | 67.32 | 52.81 | 65.91 | 50.72 | **70.36** |
| mAcc | 61.77 | 47.80 | 53.70 | 55.04 | 50.62 | **64.86** |
| mIoU | 46.37 | 33.94 | 34.66 | 41.31 | 32.62 | **49.42** |

**Implementation Details.** For the image recognition task, we divide the vision transformer backbone [53] into 3 ($D = 3$) groups, where each group contains 4 ($L = 4$) MSA-FFN modules and one multi-head interpreter. We optimize the entire framework for $D \times 30$ epochs. During every 30 epochs, we optimize the multi-head interpreter for 10 epochs and all of the subsequent MSA-FFN modules for 20 epochs. We use a mini-batch size of 32 images per GPU and adopt Adam [30] optimizer with an initial learning rate of 4e-5, which decays by cosine strategy [36] to train all our models. For the video understanding task, we set $D = 1$, i.e., we only select the informative patches at the input level. And we train the multi-head interpreter for 5 epochs and then finetune the backbone network for 1 epoch, mainly following the settings listed in the original paper [3]. We use a mini-batch size of 8

Table 2: Redundancy reduction results of our IA-RED$^2$ with DeiT on ImageNet-1k [31].

| Arch. | Method | speed (fps) | Top-1 | Top-5 | Arch. | Method | speed (fps) | Top-1 | Top-5 |
|---|---|---|---|---|---|---|---|---|---|
| | *original* | ≤930 | *79.8* | *95.0* | | *original* | ≤320 | *81.8* | *95.6* |
| | random | ≥1360 | 78.4 | 94.2 | | random | ≥440 | 80.2 | 94.6 |
| DeiT-S | MemNet | ≤350 | 77.6 | 93.6 | DeiT-B | MemNet | ≤190 | 79.9 | 94.5 |
| | attention | ≥1360 | 78.4 | 94.1 | | attention | ≥440 | 80.6 | 94.8 |
| | ours | ≥1360 | **79.1** | **94.5** | | ours | ≥440 | **80.9** | **95.0** |

video clips per GPU and adopt an SGD optimizer with an initial learning rate of 2.5e-3 in cosine strategy [36]. We train most of our models using 16 NVIDIA Tesla V100-32GB GPUs.

## 4.1 Emergence of Interpretability

In this section, we demonstrate the interpretability that emerges in our proposed method. We first show some qualitative results to show our method can better interpret where the informative region for the correct prediction is. Then we provide the quantitative results of our weakly-supervised image segmentation experiments on ImageNet which demonstrated that our method can better localize the salient object on the input images.

**Qualitative Evaluation.** We visualize the output of the multi-head interpreter in the second network group to be our results, where we choose the DeiT-S model to be the testbed. Then, we compare our method with another two baselines: (1) *Memorability map*, generated by MemNet [29], models how memorable is a certain region of the image with a concrete score ranging from zero to one. Intuitively, a memorability map highlights the region which stimulates our brains more dramatically. (2) *Raw attention*, coming inherently with the pre-trained vision transformer, highlights the regions on the image with more significant attention weight. We generate the raw attention map by averaging the attention weights between the $CLS$ token and the other patch tokens across all of the heads in the Block_1, similar to the process in Eq. 1. We demonstrate the comparison of our method with the two baselines in Figure 3, from which we can see that our proposed method localizes the objects of interest, especially the part-level stuff, more accurately. From the third example in the second column, we can see that the heatmap generated by our method combines the pattern of memorability map which detects the truck head, while the attention map highlights more irrelevant regions, such as the cloud in the sky. In the example of the shark at the sixth row of the first column, our method accurately localizes the fin and the head, the two most informative features of the shark, which indicates that our method can better interpret the part-level stuff compared to the raw attention. Also, examples of dogs and tigers at the seventh and eighth rows demonstrate that our method can detect the features on the animal face, like eyes, noses, and ears.

**Weakly-supervised Image Segmentation.** To quantitatively compare our method with other model interpretability methods, we conduct the weakly-supervised image segmentation experiments on the ImageNet-Segmentation [16] dataset. Besides the memorability map and the raw attention map, we compare with LIME [42], gradient propagation method GradCAM [44] and Layer-wise Relevance Propagation method LRP [4]. The goal of weakly-supervised ImageNet-Segmentation task is to predict a precise mask of the objects of interest without pixel-level supervision, where a binary mask served as the ground-truth label. We list the segmentation results in Table 1, from which we observe that the proposed method IA-RED$^2$ significantly outperforms other methods. As a dedicated method for the interpretability of CNNs, we find that GradCAM does not perform well in this task. We guess this is because there is a significant gap between the interpretability of CNNs and vision transformers.

## 4.2 Redundancy Reduction on Recognition Tasks

In this section, we conduct our experiments on top of the image recognition task and video action recognition task. We demonstrate that our method can hierarchically reduce the redundancy of the input patch tokens with both qualitative results and quantitative results.

**Results of Image Recognition.** We compare with different baselines for dropping the input patch tokens, such as (1) *random baseline*, which randomly drops the path tokens at the input level, (2) *MemNet baseline*, which drops the patch tokens based on the memorability score of the corresponding

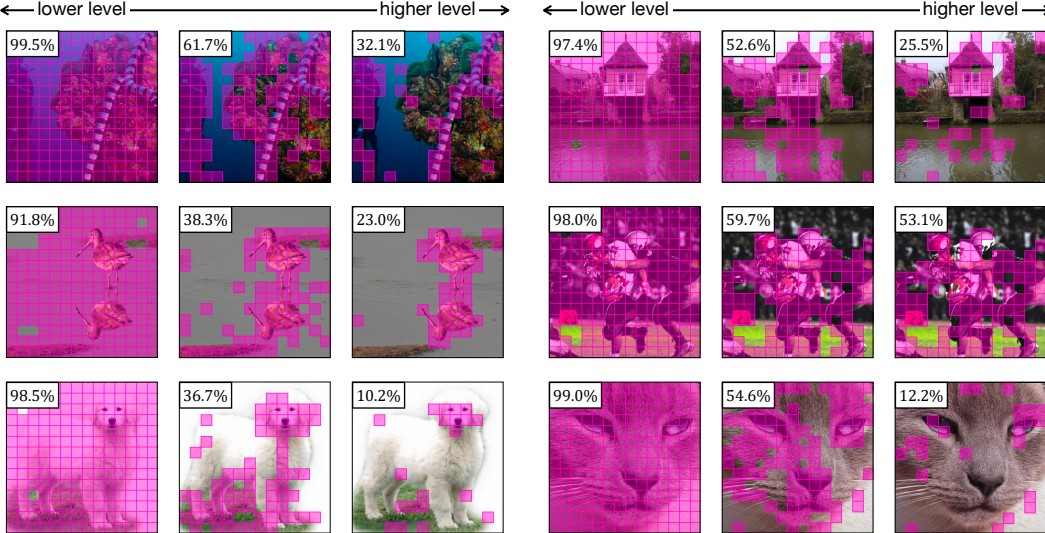

Figure 4: We visualize the hierarchical redundancy reduction process of our method with the DeiT-S model. The number on the upper-left corner of each image indicates the ratio of the remaining patches. From left to right, we can see that the network drops the redundant patches and focuses more on the high-level features of the objects. Best viewed in color.

patch, (3) *attention baseline*, which drops the patch tokens based on the raw attention map in `Block_1`. For the baseline methods, we adjust the threshold to set the drop-out rate as 30%, thus there would 30% of the patch tokens be discarded right after the positional embedding layer. Note that all of the baseline methods need re-training. For the attention baseline, since the patches are dropped according to different strategies, we fine-tune the backbone network to make it adapt to the fewer-patch case. Thus our method will not increase training time compared to the baseline methods. We choose the model of our method with the similar inference speed of (1), which would be the baseline with the fastest speed as it does not need any pre-process to the patch tokens, to fairly compare the performance. We test the inference speed in terms of frames per second (fps) of each method on a single NVIDIA Tesla V100-32GB GPU with PyTorch 1.7 and CUDA 10.2. We list our quantitative results in Table 2. In Figure 4, we visualize several examples of hierarchically dropping the patch tokens at different groups, where we can see that our model takes almost all of the tokens at the lower level of the network (the first group), then drops the patches on the background and focuses on the entire object of interest at the middle level (the second group). And finally, it focuses on the part-level stuff of the object at the higher level (the third group).

To further verify the effectiveness of our approach, we compare with a teacher-student baseline, by following the distillation process in DeiT [53]. We use DeiT-S as the teacher model and customize a student model by reducing the depth of the DeiT-S so that the student model contains only around 70% FLOPs of the DeiT-S (similar to the FLOPs of our method applied to DeiT-S). We notice that the student model achieves a top-1 accuracy of 76.0%, while our method got 79.1% top-1 accuracy on ImageNet1K with additional benefits from good model interpretability that shows what is the informative region for the correct prediction of classification.

**Results of Video Action Recognition.** We further explore the redundancy reduction strategies on the video action recognition task. Similar to the image task, we compare our method with *(1) random* baseline and *(2) attention map* baseline. Besides these two, we devise the *(3) temporal difference* baseline, which calculates the $L_2$ distance between the patch tokens at $T$ and $(T-1)$ time step. For those tokens that have a longer distance to the previous one, we assume they have larger entropy thus need to be kept. For the patch tokens sampled from the initial frame, we set their previous tokens as zero. We list the results in Table 3. We can see that our method outperforms the attention baseline while worse than the random and temporal difference baseline. Although our method does not get the best results of the four redundancy reduction methods, it learns to identify the informative patches among thousands of input patch tokens, which will be further illustrated in the the supplementary material. We guess the reason why the random baseline performs better is that the input redundancy of video is significantly higher than that in the image, which makes the model quite robust to random patch dropping as the similar technique is applied in the training process.

Table 3: Redundancy reduction results of our IA-RED$^2$ with TimeSformer on Kinetics-400 [7]. We compare three baseline methods with our IA-RED$^2$: *(1) random*, *(2) attention map*, and *(3) temporal difference*. The speed (clip per second) of each method is measure on a single NVIDIA TESLA V100-32GB GPU. For the performance, we report the clip-1 error and video-1 error, where lower is better. To be consistent, we report our reproduced results of the original TimeSformer as our Kinetics-400 dataset does not include all of the original data.

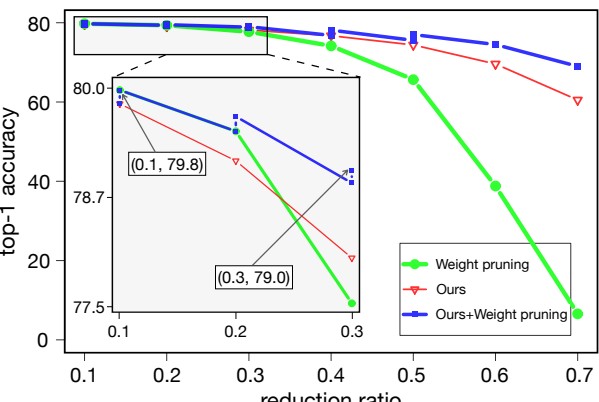

| Method | speed | clip-1 | video-1 |
|---|---|---|---|
| *original* | *≤24.0* | *28.2* | *23.8* |
| random | ≥81.0 | 34.3 | 28.2 |
| attention | ≥81.0 | 38.1 | 31.4 |
| temp. diff. | ≥81.0 | **32.3** | **26.9** |
| ours | ≥81.0 | 35.3 | 29.1 |

Figure 5: We plot the redundancy reduction results of weight pruning, our IA-RED$^2$, and the combination of them, where the $X$ axis represents the FLOPs reduction ratio of all linear layers in the DeiT-S, and $Y$ axis represents the top-1 accuracy on ImageNet. For a fair comparison, we do not finetune the network after we reduce the redundancy. With the combination of our IA-RED$^2$ and the weight pruning method, the model can be directly accelerated by $1.7\times$ **without finetuning** while suffering only a 1.7% accuracy drop.

**Comparison with Data-dependent Sparse Transformers.** We conduct additional baseline experiments by applying the Sparse Sinkhorn Attention [52] and Routing Transformer [43] to the DeiT-S model. Since the input sequential length to the DeiT-S model is fixed to 197, we set the local window size of Sinkhorn Transformer and Routing Transformer to 197. We notice that the Sinkhorn Transformer model archives 77.9% top-1 accuracy with 720 fps inference speed, while Routing Transformer achieves 77.7% top-1 accuracy on ImageNet1k and 663 fps on an NVIDIA Tesla V100 GPU. In contrast, our method obtains 79.1% top-1 accuracy with the inference speed of 1360 fps. Furthermore, we compare with Linformer [55] and observe that it only gets the top-1 accuracy of 75.7% on ImageNet1k. These results show the efficacy of IA-RED$^2$ over existing data-dependent sparse transformers in reducing the redundancy of vision transformers.

**Applicability of IA-RED$^2$.** Our approach is model-agnostic, which allows it to serve as a plugin operation for a wide range of sequence-based vision transformer architectures. Our method can be adopted to prune tokens in data-independent transformers like Swin [34], which adopt complex modifications by introducing CNN-like local windows. But since the number of tokens in different local windows will be different after sparsification, it can be hard to achieve additional speedup on top of such models. However, our model can be easily improved by using a stronger backbone to provide a better accuracy-speed trade-off compared to the Swin transformer. For example, with CaiT-S24-224 [54] as the backbone, we obtain 82.9% top-1 accuracy with only 7.5 GFLOPs (the original CaiT model got 83.5% top-1 accuracy with 9.4 GFLOPs), which is much better than the DeiT-B [53] (81.8% top-1 accuracy and 16.8 GFLOPs) and comparable to the Swin-S (83.3% top-1 accuracy with 8.7 GFLOPs) and Swin-B (83.5% top-1 accuracy with 15.4 GFLOPs). Moreover, our method for interpretability does not require fine-tuning of the original model. Results in Table 2 and 3, do not include the fine-tuning step, which is not essential in our method and is only used for getting better accuracy-speed trade-offs. Since our method does not alter the weights of the original model, it is very convenient to use as a model interpretability method for vision transformers.

**Is Data-level Redundancy Orthogonal to the Model-level?** Contrary to those works [19, 23, 35] prune the model-level redundancy, our approach seeks to reduce the data-level redundancy. To further study these two counterparts, we start by choosing the magnitude-based weight pruning approach [35] as the subject. The pruning method is applied to all of the FC layers in the transformer. We first plot the trade-off curves between accuracy and efficiency of the weight pruning and our IA-RED$^2$ in Figure 5, where we can see that our IA-RED$^2$ outperforms the weight pruning especially when the FLOPs reduction ratio is high. Then we combine these two methods to see if they are complementary to each other: for each compression step, we choose the model with higher accuracy achieved by either increasing the weight pruning ratio or lifting the threshold of our multi-head interpreter to

Table 4: Redundancy reduction results of our IA-RED$^2$ with the DeiT-Base in different resolutions. To fairly compare the ratio of the redundancy patches, we keep the parameters of MSA-FFN modules the same as the original and only optimize the multi-head interpreters.

| resolution | method | speed (fps) | FLOPs$_{avg}$ | Top-1 | speedup ratio | gap |
|---|---|---|---|---|---|---|
| 224×224 | *original* | *316.8* | *16.8 G* | *81.8* | 1.42× | 1.5% |
| | ours | 452.5 | 11.8 G | 80.3 | | |
| 384×384 | *original* | *87.0* | *49.4 G* | *82.8* | 1.48× | 0.9% |
| | ours | 129.6 | 34.7 G | 81.9 | | |

Table 5: Results of weakly-supervised image segmentation and image classification using different blocks. We use our method based on the training with DeiT-S model.

| Blocks | Weakly-Supervised Segmentation | | | Image Classification | | |
|---|---|---|---|---|---|---|
| | pixel accuracy | mAcc | mIoU | FLOPs | Top-1 | Top-5 |
| Block_0 | 68.3 | 50.5 | 37.1 | 2.9 G | 78.6 | 94.2 |
| Block_1 | 67.9 | 61.8 | 46.4 | 3.3 G | 78.4 | 94.1 |
| Block_2 | 71.9 | 55.7 | 42.6 | 3.6 G | 79.3 | 94.6 |

reduce more redundant tokens. We observe that the combined approach achieves the best trade-off, suggesting that the proposed data-level redundancy is orthogonal to the model-level redundancy, and our method is complementary to the weight pruning method.

**Does the Higher Resolution has More Redundancy?** From Figure 4, we can see the redundancy varies depending on the input instance: images with more background or small features which can identify the object tend to have more redundancy while the images with more complicated object always need more computation. Intuitively, as the redundancy is input-data-dependent, the input data with higher resolution would contain more redundancy. To validate this, we conduct the ablation study in Table 4, where we keep the reduction ratio of the computational cost the same and compare the accuracy loss of the models in different resolutions. We find that the model which takes higher resolution suffers a lower performance decrease than that with the lower resolution, which supports that input data in higher resolution tends to contain more redundancy.

**Effect of the Number of Groups D.** We conduct an ablation study on the number of groups $D = 2$, $D = 3$, and $D = 4$ on ImageNet-1K dataset. Under the same level of computational budget ($\sim$2.9 GFLOPs), we find that the 3-group framework used in our approach (79.1% top-1 accuracy) performs slightly better than the 2-group (78.6% top-1 accuracy) and the 4-group (78.8% top-1 accuracy) framework. All of them have good trade-offs between accuracy and speed.

**Effect of Different Blocks in Raw Attention Baseline.** We provide both the segmentation results and classification results of `Block_0`, `Block_1`, and `Block_2` in Table 5. From the segmentation results, we can see that the attention map of `Block_1` outperforms `Block_0` and `Block_2` by a large margin in terms of mean accuracy and mean IoU. That's why we chose `Block_1` as our baseline. From the classification results, we can see that `Block_1` and `Block_0` have similar performance on the classification task on ImageNet-1k. However, `Block_1` suffers a slightly higher computational cost. `Block_2` performs the best, but to get the attention map of `Block_2`, we need to forward two full blocks of the original vision transformer, which introduces additional computational cost. Additional results and analysis including more visualizations are included in the supplementary material.

## 5    Conclusions

In this work, we propose a novel interpretability-aware redundancy reduction framework for the recent vision transformer, named IA-RED$^2$ . We show that IA-RED$^2$ hierarchically reduces the computational cost and speeds up the vision transformer effectively with human-understandable trajectories. Experiments are conducted on image classification and video understanding tasks, where the proposed IA-RED$^2$ is demonstrated to be both model-agnostic and task-agnostic. We finally compare our IA-RED$^2$ with the model compression approaches, such as weight pruning, to demonstrate the complementarity between them.

**Acknowledgements.** We thank IBM for the donation to MIT of the Satori GPU cluster. This work is supported by the MIT-IBM Watson AI Lab and its member companies, Nexplore and Woodside.

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
