# IA-RED$^2$: Interpretability-Aware Redundancy Reduction for Vision Transformers (Supplementary Material)

**Bowen Pan[1], Rameswar Panda[2], Yifan Jiang[3], Zhangyang Wang[3], Rogerio Feris[2], Aude Oliva[1,2]**
[1]MIT CSAIL, [2]MIT-IBM Watson AI Lab, [3]UT Austin

Project Page: `http://people.csail.mit.edu/bpan/ia-red/`

## 1 Pseudo Code of Our Training Process

To make our training process clearer, we present the details of our training process by pseudo code in Alg. 1. We take the training of DeiT-S for example, where the $D = 3$. For each groups, we spend 10 epochs to train the multi-head interpreter and then 20 epochs to train the rest MSA-FFN blocks.

## 2 Discussion on the Training Time of Our Method

The 90-epoch training for DeiT-S model takes around 4.5 hours using 24 NVIDIA Tesla V100-32GB GPUs. For one third of all the epochs, we train the multi-head interpreters using REINFORCE, which does not require gradients for the backbone network and saves a lot of computation.

## 3 Random Baseline with Different Seeds

To understand how different seeds affect the experiment results, we provide the results of random dropping and dropping with our learned policy with DeiT-S using four random seeds in the table below. We can see that our method consistently outperforms the random baseline with different seeds.

Table 1: The performance of the random baseline and our method with different seeds.

| method | Top-1 (s1) | Top-1 (s2) | Top-1 (s3) | Top-1 (s4) | Average Top-1 |
|--------|-----------|-----------|-----------|-----------|---------------|
| random | 78.4% | 78.3% | 78.5% | 78.3% | 78.4% |
| Ours | 79.1% | 78.8% | 79.0% | 79.2% | 79.0% |

## 4 REINFORCE vs. Straight-through Gumbel

We also explore training with straight-through Gumbel instead of REINFORCE to be part of our approach. However, we find that Gumbel does not consistently highlight the informative region. In most cases, it highlights the background region instead of foreground objects. Here we provide the comparison of the REINFORCE method and straight-through Gumbel method on the DeiT-S model. Under the same level of FLOPs of 3.0G (Gumbel) versus 2.9G (REINFORCE), the Top-1 accuracy on ImageNet-1K dataset are 78.8% and 79.1% respectively. The results of the Gumbel method are obtained by discarding the patch tokens which have relatively higher softmax value due to the fact that, in Gumbel method, the background region tends to have higher softmax value.

35th Conference on Neural Information Processing Systems (NeurIPS 2021).

**Algorithm 1** Optimize multi-head interpreters and MSA-FFN blocks on DeiT-S.

---

**Require:** A token sequence $X$ right after the positional embedding and its label $Y$.
   **for** $i \leftarrow 1$ to $D$ **do**
      **for** $j \leftarrow 1$ to 10 **do**
         **for** each iteration **do**
            $R \leftarrow \texttt{Reward}(X,\ Y\ |\ W_p^{1:i},\ W_b)$
            $\texttt{Compute\_Policy\_Gradient}(R)$
            $W_p^i \leftarrow \texttt{Update\_Parameters}(W_p^i)$
         **end for**
      **end for**
      **for** $j \leftarrow 11$ to 30 **do**
         **for** each iteration **do**
             $L \leftarrow \texttt{CrossEntropyLoss}(X, Y \,|\, W_p^{1:i}, W_b)$
             $\texttt{Compute\_Gradient}(L)$
             $W_b^{i:D} \leftarrow \texttt{Updated\_Parameters}(W_b^{i:D})$
         **end for**
      **end for**
   **end for**
   where $D$ is the number of the groups we defined in Section 3 of the main paper, $W_p$ denotes the parameters of the multi-head interpreters, $W_b$ denotes the parameters of the MSA-FFN blocks.

---

## 5 Effect of Threshold in Discarding Tokens

We vary the threshold of $I_{i,j}$ to 0.48, 0.49, 0.50, 0.51, and 0.52, to see how the performance of the DeiT-B model would change. The results are shown in Table 2, where we find that with a higher threshold, we get a more efficient model. While lowering the threshold, we get a more accurate model. Thus the threshold of $I_{i,j}$ can be regarded as a trade-off factor between accuracy and efficiency.

Table 2: The performance of the DeiT-B model with different thresholds in discarding tokens.

| Threshold | 0.48 | 0.49 | 0.50 | 0.51 | 0.52 |
|---|---|---|---|---|---|
| FLOPs | 16.5 G | 15.3 G | 11.8 G | 8.2 G | 4.9 G |
| Top-1 | 81.7% | 81.5% | 80.9% | 77.5% | 63.7% |

## 6 Ablation Study on Square Reward and Insights on $\tau$

We jointly study the effect of replacing the squared reward with linear and changing the value of $\tau$ in Eq. 2 in the table below. We can see from that table that, by changing $\tau$ we can get different trade-offs between accuracy and efficiency. Also, without squared reward, we can see that the accuracy-efficiency trade-offs will be more sensitive to the changing of the $\tau$.

Table 3: The effect of square reward and different $\tau$.

| $\tau$ | 0.5 | 1.0 | 1.5 | 0.5 | 1.0 | 1.5 |
|---|---|---|---|---|---|---|
| square reward | Yes | Yes | Yes | No | No | No |
| Top-1 | 76.0% | 78.1% | 79.1% | 12.0% | 70.9% | 78.2% |
| FLOPs | 2.5 G | 2.9 G | 3.4 G | 0.4 G | 2.2 G | 3.1 G |

## 7 More Interpretability Results and Demo Tool

In this section, we present more visualization results on both image and video tasks. We plot more interpretability results of our method in Figure 1. Then, we show more examples of hierarchical redundancy reduction process in Figure 2. Finally, in Figure 3, we visualize the input redundancy

original memorability attention  ours    original memorability attention  ours

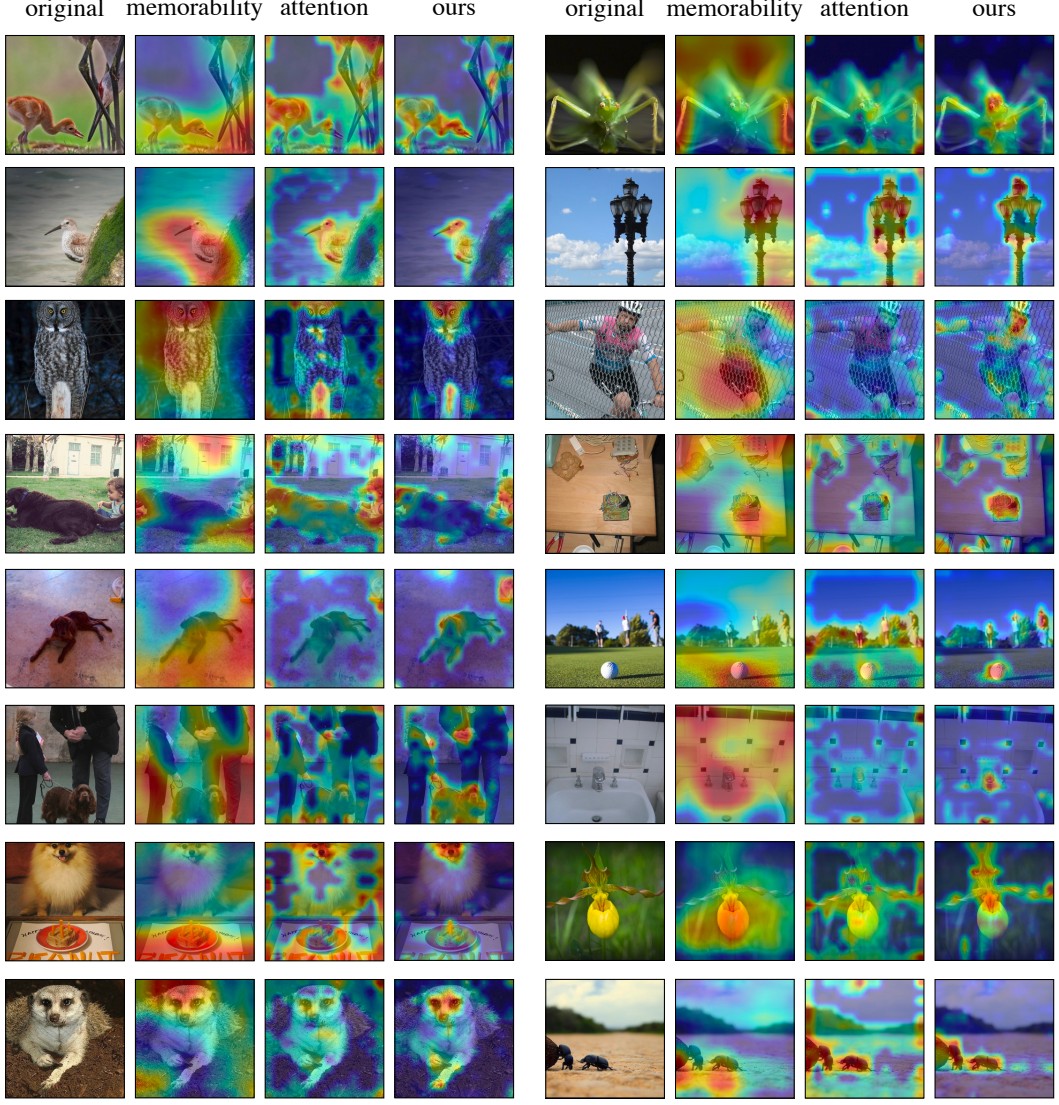

Figure 1: We visualize more examples with heatmaps which hightlight the informative region of the input images of MemNet, raw attention at the second block, and our method with DeiT-S model.

reduction results of our method on the video action recognition task, where we experiments with the JointST TimeSformer [1] on the Kinetics-400 dataset [2].

We further provide an **interpretation tool** for the reader who want to play the interpretability of our model. The usage of the tool is quite simple: `python interpreter.py -p {image_path} -o {output_dir}`. An environment with `Python==3.6` (or above), `torch==1.7` (or above) and `timm==0.3.2` (or above) installed is required to run the tool.

# 8   Broader Impact

Our work eases the suffering of heavy computational cost for the vision transformer, which could save more energy and reduce the carbon emissions for the industry. The interpretability which emerges in our method help we human to understand what happening inside the vision transformer. However, the potential negative impact would be that, since our method makes neural networks easier to run and more understandable to everyone, it may cause the abuse of AI technology.

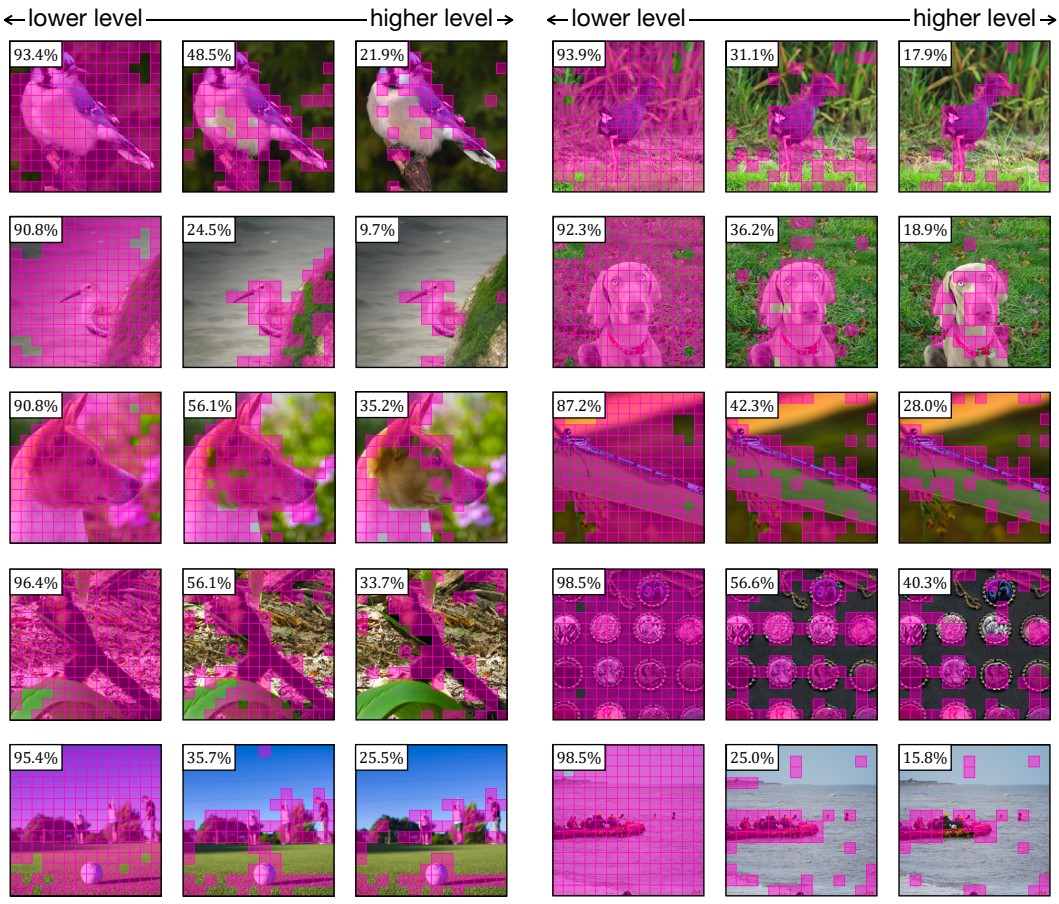

Figure 2: More examples of our hierarchical redundancy reduction process of our method with DeiT-S model. The number on the upper-left corner of each image indicates the ratio of the remaining patches. Best viewed in color.

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

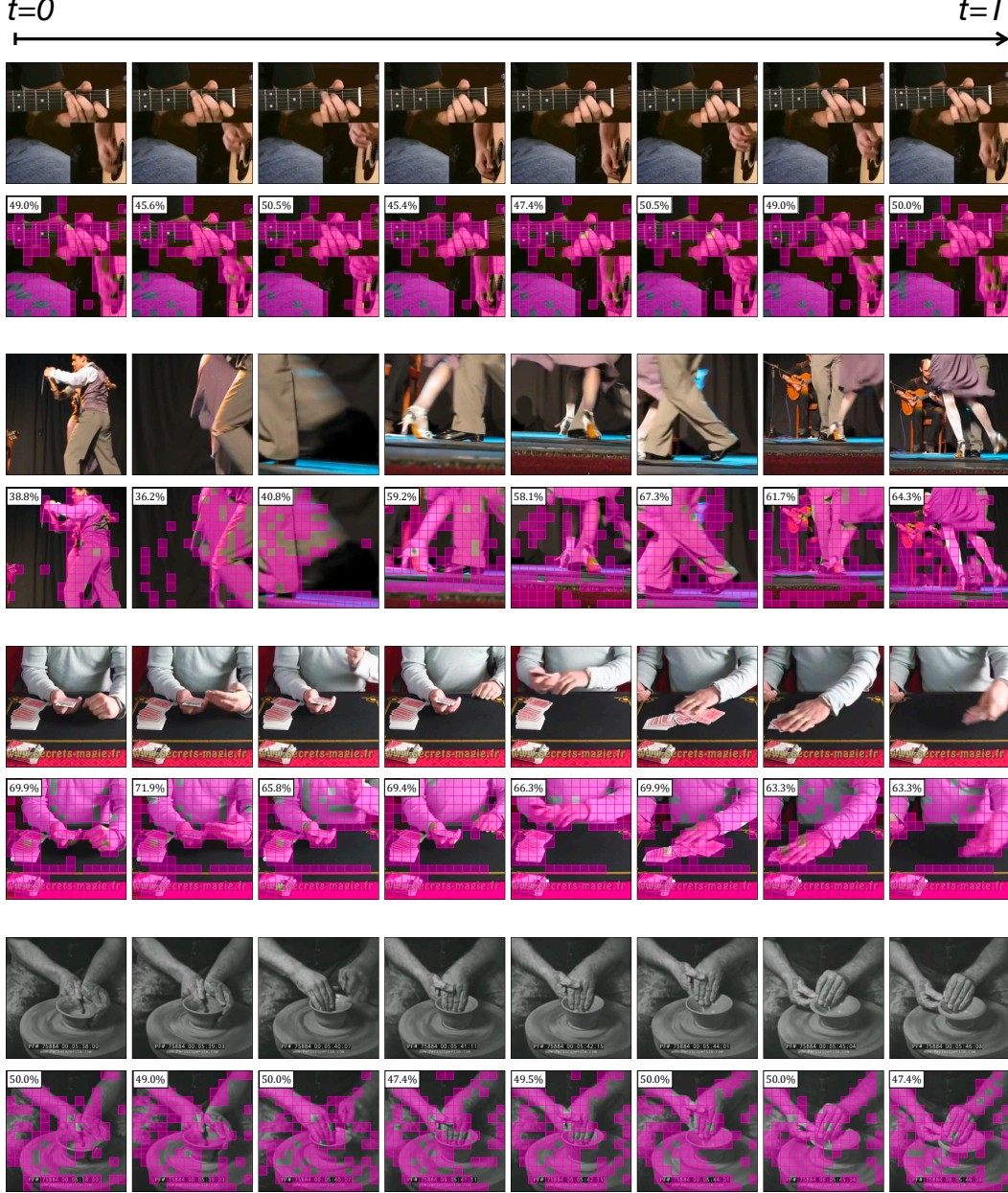

Figure 3: We visualize the redundancy results of our method with the TimeSformer model. The number on the upper-left corner of each image indicates the ratio of the remaining patches. We can see that our method manages to filter the redundant patches and keeps the informative patches which are important for the final prediction.