# OpenReview forum: "IA-RED$^2$: Interpretability-Aware Redundancy Reduction for Vision Transformers"
_NeurIPS.cc/2021/Conference — NeurIPS 2021 Poster_

### Official Review · Reviewer_yHQu · 2021-07-19

**Rating:** 4
**Confidence:** 3

**Summary:**

This paper proposes the IA-RED2 framework, which tries to reduce the redundant tokens in vision transformer in a sequential manner. The redundant tokens are determined by the proposed multi-head interpreter (as a policy network). The multi-head interpreter is trained with REINFORCE, utilizing a reward function that considers the sparsity/accuracy tradeoff. The training of the full network follows a complicated curriculum learning training pipeline, which contains multiple stages and first training multi-head interpreter and then finetuning the rest of the networks in each stage. The authors conduct experiments on ImageNet1k classification with DeiT models and Video action recognition with the TimeSformer model.

This paper is among the first wave of sparse vision transformers with data-dependent sparsity, together with several other concurrent works like "Not All Images are Worth 16x16 Words: Dynamic Vision Transformers with Adaptive Sequence Length". It attains promising interpretable results, as the authors claimed.

**Limitations And Societal Impact:**

Yes

**Main Review:**

As a CV practioner, my main concern of this work is how I can use it? If I use it as a new sparse vision transformer model (after finetuning), its accuracy-speed trade-off is not comparable with data-independent sparse transformers, like Swin Transformer. For example, Swin_Tiny can reach ImageNet top1 accuracy 81.3 with fps >= 700, which in Table 2, the authors use IA-RED finetuned DeiT-B to achieve 80.9 top1 accuracy, with fps about 440. If I use it as a model interpretability method, then IA-RED2 fine-tuned the original model for the visualizations in  Figure 3 (the scores are from the second group). Moreover, for visualization/interpretability performance, IA-RED2 is only compared with GradCAM, not other popular methods like LIME and gradient integration methods.

Therefore, if the authors want to focus on a new sparse vision transformer model, the authors may want to compare with other data-independent sparse vision transformers (HanoNet are published, Swin/ViL are are published). In fact, there are several data-dependent sparse transformers from the NLP literature, such as "Reformer: The efficient transformer", "routing transformers", "Sparse sinkhorn attention", "Informer: Beyond efficient transformer for long sequence time-series forecasting". These data-dependent sparse transformers are also model-agnostic and task-agnostic as the authors claimed, and may be good baselines for IA-RED2.

If the authors want to focus on interpretability, then the authors may want to compare with more interpretability methods (other than GradCAM) and other data-dependent sparse transformers (they are model-agnostic and task-agnostic although from NLP community), to validate the superiority of IA-RED2.

On both sides, other existing model-agnostic and task-agnostic methods from NLP community are valuable baselines to compare with.

Minor comments:

1. In Equation (2), have the authors try the reward without the square? Just 1 - |u|_0/N. The complexity of vanilla self-attention is O(12ND^2 + 2N^2D), has both linear and quadratic part. In the DeiT case (N=196), the linear part in fact even larger than the quadratic part.

2. In Equation (2), how did the authors determine the parameter \tau? Any insights here?

3. Line 241-242: "We generate the raw attention map by extracting the attention weight at the multi-head self attention module in the block 1." Since the attention is multi-head, the authors may want to elaborate on how they get the final visualization (how to aggregate attention scores from different heads).



**Time Spent Reviewing:**

5

---

> ### Author Response · Authors · 2021-08-11
> **Response to Reviewer yHQu**
>
> We would like to thank Reviewer yHQu for these valuable comments.
>
> **(a) Applicability of IA-RED^2:**  Our method is model-agnostic, which allows this to be served as a plugin operation for a wide range of sequence-based vision transformer architectures. Our method can be adopted to prune tokens in data-independent transformers like Swin, which adopt complex modifications by introducing CNN-like local windows. But since the number of tokens in different local windows will be different after sparsification, it can be hard to achieve additional speedup on top of such models. However, our model can be easily improved by using a stronger backbone to provide a better accuracy-speed trade-off compared to the Swin transformer. For example, with CaiT-S24-224 [1] as the backbone, we obtain 82.9% top-1 accuracy with only 7.5 GB FLOPs (the original CaiT model got 83.5% top-1 accuracy with 9.4 GB FLOPs), which is much better than the DeiT-B (81.8% top-1 accuracy and 16.8 GB FLOPs) and comparable to the Swin-S (83.3 % top-1 accuracy with 8.7 GB FLOPs) and Swin-B (83.5% top-1 accuracy with 15.4 GB FLOPs). We will include a discussion on this in the final version.
>
> Moreover, note that our method for interpretability does not require fine-tuning of the original model. Most of our experimental results, except Table 2 and 4, do not include the fine-tuning step, which is not essential in our method and is only used for getting better accuracy-speed trade-offs. Since our method does not alter the weights of the original model, it is very convenient to use as a model interpretability method for vision transformers.
>
> **(b) Comparison with other interpretability methods (LIME [2]):** Thanks Reviewer yHQu for pointing out these related interpretability methods which will be a good supplement to our work. Here we compare our method with LIME [2] in the weakly-supervised segmentation task and find that our method still outperforms LIME by a large margin on the ImageNet-Segmentation dataset.
>
> |      | Pixel Acc.| mean Acc. | mean IoU |
> | ----- | ----- | ----- | ----- |
> | LIME | 67.32% | 47.80% | 33.94% |
> | Ours | 70.36% | 64.86% | 49.42% |
>
> **(c) Comparison with data-dependent sparse transformers:** We thank Reviewer yHQu for providing the data-dependent literature from the NLP field. We first conduct a baseline experiment by applying the sparse sinkhorn attention [3] to the DeiT-S model. Since the input sequential length to the DeiT-S model is fixed to 197, we set the local window size of SinkhornTransformer to 197. We notice that the SinkhornTransformer model archives 77.9% top-1 accuracy on ImageNet1K and 720 fps on an NVIDIA Tesla V100 GPU, while our method obtains 79.1% top-1 accuracy with the inference speed of 1360 fps. Furthermore, we compare with Linformer [4], another efficient attention baseline from NLP literature, into comparison and observe that it only gets the top-1 accuracy of 75.7% on ImageNet1K. These results show the efficacy of our method over existing data-dependent sparse transformers (from NLP literature) in reducing the redundancy of vision transformers.
>
> **(d) Ablation study on square reward and insights on tau (Eq. 2):** We jointly studied the effect of replacing the squared reward with linear and changing the value of tau in the table below. We can see from that table that, by changing tau we can get different trade-offs between accuracy and efficiency. Also, without squared reward, we can see that the accuracy-efficiency trade-offs will be more sensitive to the changing of the tau.
>
> |  tau | squared reward? | Top-1 (no fine-tuning) | FLOPs |
> | ----- | ----- | ----- | ----- |
> | 0.5 | Yes | 76.0% | 2.5 G |
> | 1.0 | Yes | 78.1% | 2.9 G |
> | 1.5 | Yes | 79.1% | 3.4 G |
> | 0.5 | No | 12.0% | 0.4 G |
> | 1.0 | No | 70.9% | 2.2 G |
> | 1.5 | No | 78.2% | 3.1 G |
>
> **(e) Details on raw attention visualization (Line 241-242):** Thanks for pointing it out! We generate the raw attention map by averaging the attention weights between the CLS token and the other patch tokens across all of the heads, similar to the process of Eq. (1). We will elaborate on this part to make this process clearer to the readers in our final version.
>
> **References:**
>
> - [1] Hugo Touvron, Matthieu Cord, Alexandre Sablayrolles, Gabriel Synnaeve, Hervé Jégou. Going deeper with Image Transformers, arXiv preprint arXiv:2103.17239, 2021.
> - [2] Marco Tulio Ribeiro, Sameer Singh, Carlos Guestrin. "Why should i trust you?" Explaining the predictions of any classifier, ACM SIGKDD international conference on knowledge discovery and data mining, 2016.
> - [3] Yi Tay, Dara Bahr, Liu Yang, Donald Metzler, Da-Cheng Juan. Sparse Sinkhorn Attention, ICML, 2020.
> - [4] Sinong Wang, Belinda Z. Li, Madian Khabsa, Han Fang, Hao Ma. Linformer: Self-attention with linear complexity, arXiv preprint arXiv:2006.04768, 2020.

---

> > ### Author Response · Authors · 2021-08-12
> > **Experimental results on RoutingTransformer [ref. (c) Comparison with data-dependent sparse transformers]**
> >
> > As suggested by Reviewer yHQu, we just got the results on comparison with another data-dependent sparse transformer, RoutingTransformer [5]. Similar to the Sinkhorn transformer,  we applied it to the DeiT-S model and noticed that the RoutingTransformer achieves 77.7% top-1 accuracy on ImageNet1K and 663 fps inference speed on an NVIDIA Tesla V100 GPU, while our method obtains 79.1% top-1 accuracy with the inference speed of 1360 fps. We can see that our method still outperforms [5] in terms of both speed and accuracy. We will add these results in our final version.
> >
> > **Reference:**
> >
> > - [5] Aurko Roy, Mohammad Saffar, Ashish Vaswani, David Grangier. Efficient Content-Based Sparse Attention with Routing Transformers, TACL 2020.

---

> > > ### Author Response · Authors · 2021-08-19
> > > **Request for feedback on the rebuttal**
> > >
> > > Dear Reviewer yHQu,
> > >
> > > We thank the reviewer's time for reviewing, and we really hope to have a further discussion with the reviewer yHQu to see if our response solves the concerns. We have addressed all the thoughtful questions and suggestions raised by the reviewer (e.g., new comparisons with data-dependent sparse transformers and other interpretability method), and we hope that the work's impact and results are better highlighted with our responses. It would be great if the reviewer can kindly check our responses and provide feedback with further questions/concerns (if any). We would be more than happy to address them. Thank you!
> > >
> > > Best wishes,
> > >
> > > Authors

---

> > > > ### Author Response · Authors · 2021-09-01
> > > > **Request for feedback**
> > > >
> > > > Dear Reviewer yHQu,
> > > >
> > > > Thanks again for your constructive comments and suggestions. As the discussion phase is nearing its end, we wondered if you might still have any concerns that we could address. We hope our new results and comparisons with other sparse transformers and interpretability methods addressed all your questions/concerns. Thank you!
> > > >
> > > > Best wishes,
> > > >
> > > > Authors

---

### Official Review · Reviewer_goz9 · 2021-07-19

**Rating:** 6
**Confidence:** 4

**Summary:**

This work presents a novel framework for hierarchical redundancy reduction in input tokens to vision transformers, resulting in fewer computation in multi head self-attention layers and improved inference time. The method achieves 1.4x speed up compared to DeiT or TimeSformer models when applied to image recognition and video action recognition tasks. The proposed method is also more interpretable compared to other baselines and raw attention learned by original vision transformer model.

**Limitations And Societal Impact:**

Yes

**Main Review:**

STRENGTHS
- The approach reduces the number of tokens in a learnable manner, reducing the speed of inference considerably compared to models like DeiT or TimeSformer.
- The authors added good visualizations how hierarchical redundancy reduction process works.
- Interpretability results shown with the method appears to focus more on the informative regions compared to baseline methods.

WEAKNESSES AND REMARKS

- Increased training time considerably? Given that the models are initialized from pretrained weights, and then again trained for ~90 epochs, this approach is increasing the training time considerably. The gains from this approach seems to be small over an attention only baseline(which doesn't need any retraining?). For image classification, this approach is better than an attention baseline by 0.4 and for video classification, this approach is actually worse than an attention only or random baseline by a large margin.

- Given the different baselines and the IA-RED$^2$ approach give close performance results, it will be good to understand how different seeds affect the experiment results.

- The networks seem to be initialized from pretrained weights. How does this method work when training a model from scratch? If the method needs the model to be initialized from pretrained weights, is this approach better than training a student network containing fewer number of parameters OR having similar FLOPs as IA-RED$^2$ model distilled from a teacher Deit-B etc?

- Missing several important ablations for various hyper-parameters used in the methods. To understand how the hierarchical approach works, it would have been interesting to see the effect of changing the number of groups $D$. In addition, whenever $I_{i,j} < 0.5$ , the token is discarded for all the subsequent groups. How does different values for this threshold effect the final performance?

- The interpretable results in Figure 3 seems to be mixed (with raw attention attending to better regions in a few compared to IA-RED$^2$). Hence in some cases the message is not very clear. Are the raw attention results shown use all the tokens? Are the visualized results randomly chosen?

**Time Spent Reviewing:**

4

---

> ### Author Response · Authors · 2021-08-11
> **Response to Reviewer goz9**
>
> We would like to thank Reviewer goz9 for acknowledging that our method reduces the number of tokens, improves the efficiency of the transformer models, and has a good visualization of how the hierarchical redundancy reduction process works.
>
> **(a) Training time:** Our DeiT-S model for 90 epoch training takes around 4.5 hours using 24 NVIDIA Tesla V100-32GB GPUs. Moreover, for 30 epochs, we are training the multi-head interpreters in the REINFORCE manner, which doesn’t require gradients for the backbone network and saves a lot of computation.
>
> **(b) Performance improvements over raw attention baseline:** The attention baseline also needs re-training. For the attention baseline, since the patches are dropped according to the raw attention map, we fine-tune the backbone network to make it adapt to the fewer-patch case.  Thus our method won’t increase training time compared to the attention baseline. We will make it clear in the final version. Furthermore, on the video classification experiments, our approach is in fact significantly better than the attention baseline by a large margin (instead of lower performance as stated by the reviewer). Since we reported the clip-1 error and video-1 error following the metrics in the TimeSFormer paper [1], our method outperforms the attention baseline by 2.3% on video-1 error (29.1% versus 31.4%).
>
> **(c) Results with different seeds:** We provide the results of random dropping and dropping with our learned policy with DeiT-S using four random seeds in the table below. We can see that our method consistently outperforms the random baseline with different seeds.
>
> |      | Top-1 (s1) | Top-1 (s2) | Top-1 (s3) | Top-1 (s4) | Average top-1 |
> | ----- | ----- | ----- | ----- | ----- | ----- |
> | Random  | 78.4%  | 78.3% | 78.5% | 78.3% | 78.4% |
> | Ours | 79.1% | 78.8% | 79.0% | 79.2% | 79.0% |
>
> **(d) Initialization and comparison with teacher-student baseline:** We would like to thank Reviewer goz9 for this great suggestion! We observe that optimizing for both accuracy and efficiency is not effective when training our model from scratch. Specifically, we find that jointly training our multi-head interpreter and the backbone network from scratch leads to much lower performance as neither of the multi-head interpreter and the backbone network are able to learn good features. As suggested by the reviewer, we compare with a teacher-student baseline, by following the distillation process in DeiT [2]. We use DeiT-S as the teacher model and customize a student model by reducing the depth of the DeiT-S so that the student model contains only around 70% FLOPs of the DeiT-S (similar to the FLOPs of our method applied to DeiT-S). We notice that the student model achieves a top-1 accuracy of 76.0%, while our method got 79.1% top-1 accuracy on ImageNet1K. Moreover, our method benefits from good model interpretability that shows what is the informative region for the correct prediction of classification (Section 4.1). We will add a discussion on the teacher-student baseline in the final version.
>
> **(e) Effect of the number of groups D:** Thanks for this valuable suggestion. We conducted an ablation study on the number of groups D=2, D=3, and D=4 on ImageNet1K. Under the same level of computational budget (~2.9 GB FLOPs), the 3-group framework used in our approach (79.1% top-1 accuracy) performs slightly better than the 2-group (78.6% top-1 accuracy) and the 4-group (78.8% top-1 accuracy) framework. All of them have good trade-offs between accuracy and speed. We will add this analysis to our final version.
>
> **(f) Effect of threshold in discarding tokens:** We varied the threshold of I_{i,j} to 0.48, 0.49, 0.50, 0.51, and 0.52, to see how the performance of the DeiT-B model would change. The results can be found in the table below, where we find that with a higher threshold, we got a more efficient model. While lowering the threshold, we got a more accurate model. Thus the threshold of I_{i,j} can be regarded as a trade-off factor between accuracy and efficiency. We will add these studies to our final version.
>
> | Threshold  | 0.48 | 0.49 | 0.50 | 0.51 | 0.52 |
> | ----- | ----- | ----- | ----- | ----- | ----- |
> | FLOPs  | 16.5 G  | 15.3 G | 11.8 G | 8.2 G | 4.9 G |
> | Top-1 | 81.6% | 81.5% | 80.9% | 77.5% | 63.7% |
>
> **(g) Clarification on interpretable results (Figure 3):** Are the raw attention results shown use all the tokens?: Yes, the raw attention results shown use all the tokens. Are the visualized results randomly chosen?: Yes, they are randomly chosen. By observing the visualizations, we can find that our method gives more high-level interpretation than raw attention. Let’s take the results in the 1st column, 6th row in Figure 3 for an example. We can see that although the raw attention map highlights the most area of the foreground object, shark, our proposed method focuses on the most discriminative features that make the object a shark, the fin, and the head. For the results in the 2nd column and 7th row, we can see that our method focuses more on the eyes and nose of the dog instead of the entire face.
>
> **References:**
>
> - [1] Gedas Bertasius, Heng Wang, and Lorenzo Torresani. Is space-time attention all you need for video understanding? arXiv preprint arXiv:2102.05095, 2021.
> - [2] Hugo Touvron, Matthieu Cord, Matthijs Douze, Francisco Massa, Alexandre Sablayrolles, and Hervé Jégou. Training data-efficient image transformers & distillation through attention. arXiv preprint arXiv:2012.12877, 2020.

---

> > ### Comment · Reviewer_goz9 · 2021-09-01
> > **post rebuttal**
> >
> > Thanks to the authors for addressing all my concerns.

---

> > > ### Author Response · Authors · 2021-09-01
> > > **Thanks**
> > >
> > > Thanks for all the valuable feedback. We’re glad that our response addressed all your concerns.

---

### Official Review · Reviewer_8Ne1 · 2021-07-20

**Rating:** 7
**Confidence:** 4

**Summary:**

The paper presents a method to improve the efficiency of vision transformers. Traditional transformers process maintain and processing embeddings equal to the number of patches sampled from the original visual input. The key idea of this work is to modify the existing architectures to allow dropping of redundant embeddings at intermediate stages. The experiments demonstrate that the approach improves efficiency, retains accuracy and effectively retains relevant patches from the input signal while discarding redundant areas.

**Limitations And Societal Impact:**

The authors have adequately addressed the limitations and societal impact.


**Main Review:**

The paper is well written and is very easy to follow for the most part (see comment below). The introduction concisely motivates the problem and presents an overview of the paper. The related work section does a thorough job of contrasting with existing work in this domain.

The idea of learning an additional model to drop redundant embeddings is novel and makes an interesting contribution towards improving the efficiency of transformers in computer vision (to the best of my knowledge).

The proposed "Multi-head interpreter" architecture is intuitive and provides a simple way to implement the idea of redundancy reduction. The simplicity of this architecture makes this idea generally applicable to a wide-range of domains with minimal changes. I believe that this has the potential to be widely adopted.

The experimental evaluation is thorough and exhaustive. The qualitative visualizations show conclusive evidence that the proposed "Multi-head Interpreter" focuses on highly relevant regions of the input signal while discarding redundant regions. The quantitative evaluation further demonstrates the benefits of this approach by showing improved efficiency without any loss in performance.

The paper also presents some additional interesting insights - lower redundancy in cluttered images and orthogonality of model redundancy vs data redundancy.

## Questions, Concerns and Suggestions

- Line 158-161, "the VIT is divided into D groups, ...". Here it is not clear that these D groups are sequential components of the whole VIT. This is especially confusing since the Figure 2 shows them as parallel pipelines. This could be explained better.
- Line 162: "The network groups are optimized separately from the group right after the patch embedding layer to the group right before the prediction head." This line is not clear immediately and only makes sense after reading the next section.

- Question: Why is REINFORCE used to train the interpreter? Why is it not trained in a supervised fashion using a gradient estimator (like the straight-through gumbel)?

- Line 183-197 is very hard to follow. It would really help if a pseudo code for this training process was provided.

- For the raw attention baseline, why is the attention chosen from block 1 specifically? Does it do better or worse (for results in Table 2) if the attention is chosen from a different block?





**Time Spent Reviewing:**

3 hours

---

> ### Author Response · Authors · 2021-08-11
> **Response to Reviewer 8Ne1**
>
> We would like to thank Reviewer 8Ne1 for confirming our idea is novel, simple yet effective, and our experiments are thorough and exhaustive.
>
> **(a) Clarification on D groups (Line 158-161):** Here we mean that all of the MSA-FFN blocks in the original vision transformer will be evenly assigned into D groups in our IA-RED^2 framework, where each group contains L number of MSA-FFN blocks and one multi-head interpreter. We will rephrase this part in our final version. Thanks for pointing it out!
>
> **(b) Optimization details (Line 162):** The network groups are optimized in a curriculum learning manner. For example, if the number of groups D is 3, we will first optimize groups 1-3, then 2-3,  and finally, we optimize the third group. We agree with Reviewer 8Ne1 that this sentence is confusing. We will make it clearer in the final version with more restricted clarification and examples which are easy to understand.
>
> **(c) REINFORCE vs Straight-through Gumbel:** We did explore the supervised fashion with straight-through Gumbel to be part of our approach in the earlier phase of our research. However, according to the visualization, we found that Gumbel didn’t consistently highlight the informative region -- in most cases, it highlighted the background region instead of foreground objects. Here we provide the comparison of the REINFORCE method and straight-through Gumbel method on the DeiT-S model. Under the same level of FLOPs of 3.0 G (Gumbel) versus 2.9 G (REINFORCE), the Top-1 accuracy on ImageNet1K are 78.8% and 79.1% respectively. The results of the Gumbel method are obtained by discarding the patch tokens which have relatively higher softmax value according to our visualization. We can see that our method has advantages in both computational cost and accuracy. Moreover, our method delivers better model interpretability compared with the straight-through Gumbel method. We will include this discussion in the final version.
>
> **(d) L183-197 and Pseudo-code:** We will polish the writing of L183-197 in the final version. For the pseudo code, we actually included it in our supplementary materials (Section A) where we took the training of DeiT-S with D=3 for example. For each group, we spend 10 epochs to train the multi-head interpreter and then 20 epochs to train the MSA-FFN blocks.
>
> **(e) Results using different blocks in raw attention baseline:** Thanks for this suggestion. We provide both the segmentation results and classification results of Block_0, Block_1, and Block_2 in the table below. From the segmentation results, we can see that the attention map of Block_1 outperforms Block_0 and Block_2  by a large margin in terms of mean accuracy and mean IoU. That’s why we chose Block_1 as our baseline. From the classification results, we can see that Block_1 and Block_0 have similar performance on the classification task on ImageNet1K. However, Block_1 suffers a slightly higher computational cost. Block_2 performs the best, but to get the attention map of Block_2, we need to forward two full blocks of the original ViT, which introduces additional computational cost.
>
> **Weakly-supervised Image Segmentation on ImageNet-Segmentation dataset with the DeiT-S:**
>
> |            | Pixel Acc. | mean Acc. | mean IoU|
> | ------------- | ------------- | ------------- | ------------- |
> | Block_0  | 68.3%  | 50.5% | 37.1% |
> | Block_1  | 67.9%  | 61.8% | 46.4% |
> | Block_2  | 71.9%  | 55.7% | 42.6% |
>
> **Classification on ImageNet1K dataset with DeiT-S:**
>
> |  | FLOPs | Top-1 | Top-5 |
> | ----- | ----- | ----- | ----- |
> | Block_0  | 2.9 G  | 78.6% | 94.2% |
> | Block_1  | 3.3 G  | 78.4% | 94.1% |
> | Block_2  | 3.6 G  | 79.3% | 94.6% |

---

> > ### Author Response · Authors · 2021-09-01
> > **Request for feedback on the rebuttal**
> >
> > Dear Reviewer 8Ne1,
> >
> > Thank you for your constructive comments and suggestions. As the discussion phase is nearing its end, we wondered if you might still have any concerns that we could address. We believe our response addressed all your questions/concerns, and hope that the work's impact and results are better highlighted with our responses. Thank you!
> >
> > Best wishes,
> >
> > Authors

---

### Author Response · Authors · 2021-08-11
**Summary of Author Response**

We would like to thank all the reviewers for their constructive comments! We are glad that the reviewers found that: **(a)** our idea is novel and makes an interesting contribution towards improving the efficiency of vision transformers (Reviewer 8Ne1); **(b)** our paper is among the first wave of sparse vision transformers with data-dependent sparsity (Reviewer yHQu); **(c)** our approach is effective and has the potential to be widely adopted (Reviewer 8Ne1); **(d)** our experimental evaluation is strong with both exhaustive quantitative results (Reviewer 8Ne1, Reviewer goz9) and informative qualitative results (Reviewer 8Ne1, Reviewer goz9).

We have addressed all the questions that the reviewers posed with additional experimental comparisons and clarifications. All of these additional experiments and suggestions will be included in the final version.

---

### Author Response · Authors · 2021-08-27
**Feedback on the rebuttal**

Dear Reviewers,

We have addressed all the questions raised by the reviewers in the individual responses below. It would be great if the reviewers can kindly check our responses and provide feedback with further questions/concerns (if any). We would be more than happy to address them. Thank you!

Best wishes,

Authors

---

### Decision · Program_Chairs · 2021-09-27

**Decision:**

Accept (Poster)

**Comment:**

This paper proposes a framework for redundancy reduction in vision transformers. The reduction is obtained in a learnable manner, using REINFORCE. The results are encouraging, yielding an improvement over recent vision transformer architectures and providing an interpretable result. The tackled topic is timely and the execution of this submission is good. Therefore, I recommend the acceptance of this paper as a poster.